# Characteristics of Inorganic Phosphate-Solubilizing Bacteria from the Sediments of a Eutrophic Lake

**DOI:** 10.3390/ijerph16122141

**Published:** 2019-06-17

**Authors:** Yong Li, Jiejie Zhang, Jianqiang Zhang, Wenlai Xu, Zishen Mou

**Affiliations:** 1Faculty of Geosciences and Environmental Engineering, Southwest Jiaotong University, Chengdu 610059, China; txgsfy@163.com (J.Z.); zhjiqicn@swjtu.cn (J.Z.); 2State Key Laboratory of Geohazard Prevention and Geoenvironment Protection, Chengdu University of Technology, Chengdu 610059, China; mouzishen17@cdut.edu.cn; 3Haitian Water Grp Co. Ltd., Chengdu 610059, China

**Keywords:** inorganic phosphate-solubilizing bacteria (IPB), diversity, HCl-P, sediment, P release

## Abstract

Inorganic phosphate-solubilizing bacteria (IPB) are an important component of microbial populations in lake sediments. The phosphate that they decompose and release becomes an important source of phosphorus for eutrophic algae. The IPB strains were screened and isolated from the sediments of Sancha Lake using National Botanical Research Institute’s phosphate (NBRIP) plates. Their taxonomy was further determined by the 16S rDNA technique. The tricalcium phosphate-solubilizing ability of obtained IPB strains was evaluated using NBRIP- bromophenol blue (BPB) plates and Pikovskaya (PVK) liquid medium. Then, the ability of IPB strains to release phosphorus from the sediments were investigated by mimicking the lake environment. In this study, a total of 43 IPB strains were screened and isolated from the sediments of Sancha Lake, belonging to three phyla, eight families, and ten genera. Among them, two potentially new strains, SWSI1728 and SWSI1734, belonged to genus Bacillus, and a potentially new strain, SWSI1719, belonged to family Micromonosporaceae. Overall, the IBP strains were highly diverse and Bacillus and Paenibacillus were the dominant genera. In the tricalcium phosphate-solubilizing experiment, only 30 of the 43 IPB strains exhibited clear halo zones on plates, while in the liquid culture experiment, all strains were able to dissolve tricalcium phosphate. The phosphate-solubilizing abilities of the strains varied significantly, and the strain SWSI1725 of the *Bacillus* genus showed the strongest ability with a phosphate-solubilizing content of 103.57 mg/L. The sterilized systems demonstrated significantly elevated phosphorus hydrochloride (HCl–P) decomposition and release from the sediments after the inoculation of IPB strains, whereas no significant effect was demonstrated on the phosphonium hydroxide (NaOH-P). Thus, the IPB strains in the sediments of Sancha Lake possessed rich diversity and the ability to release phosphorus in sediments.

## 1. Introduction

While assessing the exogenous phosphorus in effectively controlled eutrophic water, the release of endogenous phosphorus in sediments plays an important role at elevated phosphorus concentrations in the overlying waters of lakes [1]. Therefore, it is particularly reasonable to study the release capability and impact factors of endogenous phosphorus in sediments. According to the Standards Measurements and Testing Program of the European Commission (SMT) [2], the total phosphorus (TP) in sediments consists of organic phosphorus (OP) and inorganic phosphorus (IP), while the inorganic phosphorus (IP) consists of phosphorus hydrochloride (HCl-P, primarily calcium-bound phosphorus) and phosphonium hydroxide (NaOH-P, primarily iron/aluminum bound phosphorus). Compared to the physical and chemical effects, microorganisms have a significant impact on the release of endogenous phosphorus [3]. Inorganic phosphate-solubilizing bacteria (IPB) are an important microbial class in sediments, playing an essential role in the phosphorus cycle of eutrophic water. They produce organic acids in cells, and these organic acids are secreted from periplasm into the in vitro environment to dissolve water-insoluble inorganic phosphate (ISIP) and to generate water-soluble inorganic phosphorus (WSIP), leading to water eutrophication [4]. Until now, 39 genera and 89 species of pure-cultured IPB have been reported [5,6] although the studies were focused on the soil-crop rhizospheric IPB for improving the utilization of phosphate fertilizer in basic crops or cash crops [7].

Studies have been conducted on the numbers, types, and distributions of phosphate-dissolving bacteria in ocean, sea, river, and lake environments [4,8,9]. Likewise, there are reports on the involvement of heterotrophic microorganisms in the phosphorus cycle in freshwater and marine environments [10,11]. However, there is very limited research on the IPB strains, their characteristics, and their release potential for sediment phosphorus in a eutrophic aquatic ecosystem with phosphorus as the limiting factor and with HCl-P as the primary form of phosphate (P). Sancha Lake is a prominent source of drinking water in Jianyang City, Sichuan Province [12]. With more and more serious eutrophication, the safety of drinking water is threatened. In our previous studies, we showed Sancha Lake is a phosphorus-restricted lake, with the primary form of TP being HCl-P [13]. On controlling the external load, the release of phosphorus in the sediments might become an important source of phosphorus in the overlying water. Therefore, we proposed that a wide variety of IPB strains are able to release HCl-P from sediments and can be screened and isolated from the sediments of Sancha Lake. For this, we screened and isolated IPB strains from the sediments of Sancha Lake, carried out their classification, and investigated their ability to release phosphorus in the medium and in the sediments. Our study is essential for understanding the phosphorus release rule and for controlling the eutrophication of Sancha Lake.

## 2. Materials and Methods

### 2.1. Overview of Sancha Lake and Collection of Sediments

Sancha Lake is located in Tianfu New District, Sichuan Province, China, 30°13′08″–30°19′56″ N, and 104°11′16″–104°17′16″ E, with an average water depth of 8.3 m and a maximum water depth of 32.5 m. It has a humid mid-subtropical monsoon climate, with an average annual temperature of 15.2~16.9 °C and an average annual precipitation of 786.5 mm. Sancha Lake is important not only as a source of water in Jianyang City but also in maintaining biodiversity, water storage and irrigation, surface runoff, and climate regulations. Its primary source of water is the Min River, accounting for 80% of the total water volume of the reservoir, and the remaining 20% is fulfilled by natural rainfall and two streams [12]. According to the lake characteristics and human activities, Sancha Lake is divided into five functional areas, as shown in Figure 1, which are (I) the main lake headwater area, (II) the highly concentrated area of the original cage culture, (III) the neighboring area with concentrated human activities, (IV) the relatively concentrated area of the pen culture, and (IV) the reservoir tailwater area [13].

In April 2017, the surface sediments of the lake bed were collected from the abovementioned five functional areas of Sancha Lake (Figure 1) using a Peterson grab and sealed in polyethylene bags. Three parallel samples from each sampling site were simultaneously collected and mixed as the representative sample of that area, immediately kept in ice, and transported to the laboratory at the earliest possible time. Then, a portion of each sample was stored at approx. 20 °C for future use, and another portion was subject to subsequent experiments directly. Simultaneously, at each sampling site, a gas-tight water sampler was used to collect the overlying water on the sediment surface. The water samples were transported to the laboratory as soon as possible as per the relevant requirements for water preservation and transportation of the Regulation for Water Environmental Monitoring (SL 219-98), then filtered through a 0.22-µm membrane filter and stored at 4 °C for future use.

### 2.2. Screening and Isolation of IPB

From each fresh segment, 10 g was added to a 250-mL Erlenmeyer flask with glass beads and 100 mL of sterile 0.85% physiological saline and was shaken at 25 °C at 160 rpm for 2 h to form a sediment suspension. The suspension was diluted to make a gradient of 10^−2^, 10^−3^, 10^−4^, 10^−5^, and 10^−6^ suspension solutions. Then, with three repeats per gradient, a 0.1-mL suspension was inoculated into the National Botanical Research Institute’s phosphate (NBRIP) growth medium plate (D-glucose, 10 g; Ca_3_(PO_4_)_2_, 5 g; MgCl_2_·6H_2_O, 5 g; MgSO_4_·7H_2_O, 0.25 g; KCl, 0.2 g; (NH_4_)_2_SO_4_, 0.1 g; agar, 15 g; and distilled water, 1,000 mL, pH 7.0) [14] and cultured at 28 °C for seven days. The colonies, which had clear halo zones or were growing well, were picked, separated, and purified for five rounds. The resulting single colonies were transferred to LB (Luria-Bertani) slants (NaCl, 5.0 g; peptone, 10 g; beef extract, 3 g; agar, 15.0 g; and distilled water, 1000 mL, pH 7.0) and stored in a 4 °C refrigerator.

### 2.3. 16S rRNA Gene Amplification, Sequencing, and Phylogenetic Analysis

The bacterial total DNA was extracted using the bacterial genomic DNA isolation kit from Tiangen Biotech (Beijing, China.) Co., Ltd. as per the kit instructions. Using the extracted DNA as a template, the PCR amplification of 16S rDNA was carried out using universal primers 27F (5′- AGAGTTTGATCCTGGCTCAG -3′) and 1492R (5′- GGTTACCTTGTTACGACTT -3′) [15]. The amplification reaction was performed on a T100 Thermal Cycler (Bio-Rad Laboratories, Inc. Hercules, CA, USA). The PCR reaction system (20 µL) contained 10 × Ex Taq buffer, 2.0 µL; 5 U Ex Taq 0.2 µl, 2.5 mM dNTP Mix, 1.6 µL; primer 27F, 1 µL; primer 1492R 1 µL; template DNA, 0.5 µL; and ddH2O, 13.7 µL. The PCR conditions were as follows: an initial denaturation at 95 °C for 5 min, 25 cycles of denaturation at 95 °C for 30 s, annealing at 56 °C for 30 s, extension at 72 °C for 90 s, and a final extension at 72 °C for 10 min after the cycles. The PCR amplification products were detected on a 0.8% agarose gel by electrophoresis. The PCR products were recovered using a kit and sequenced using a 3730xl DNA Sequencer (Shanghai Majorbio Bio-pharm Technology Co., Ltd., Shanghai, China).

The raw sequencing reads were processed with SeqMan II to remove the vector and the contaminating sequences. Then, the bacterial 16S rDNA sequence data were curated to remove regions with low quality nucleotide score results (such as ambiguous bases) and then assembled into contigs. The obtained 16S rDNA full sequences were submitted to EzBioCloud (https://www.ezbiocloud.net) for a basic local alignment search tool (BLAST) analysis to determine the similarity and the species that were most closely related to the strains. For sequence editing, the effective typical similar sequences were selected from the database and compared with the obtained sequences using multiple sequence alignment from clustalx (https://www.clustal.org/). The evolutionary distance was calculated using the Kimura two-parameter model from MEGA 7.0 (https://www.megasoftware.net/), and the phylogenetic tree of the 16S rRNA sequences was constructed by the Neighbor-Joining (NJ) method, with a bootstrap value of 1000 [16].

### 2.4. Evaluation of Phosphate-Solubilizing Ability of the IPB Strains

#### 2.4.1. NBRIP Solid Culture

Each IPB strain obtained by screening and isolation was used to prepare a bacterial suspension with an OD600 value of approximately 2.5. Then, 2.0 mL of the bacterial suspension was transferred onto an NBRIP-BPB (NBRIP media with 0.025 g/L bromophenol blue (BPB)) plate invert-cultured at 28 °C for 7 days. Three plates were set up for each strain. Each colony with a clear halo zone was carefully observed, and the halo diameter (HD) and colony diameter (CD) were measured. The P-solubilizing ability of that strain was initially evaluated according to the presence or absence of the halo zone and the ratio of HD/CD. An HD/CD value of greater than or equal to 1.5 indicated a strong ability, while a value between 1.0 and 1.5 indicated a weak ability to dissolve phosphorus [17].

#### 2.4.2. PVK Liquid Culture

Each IPB strain obtained by screening and isolation was inoculated into an LB medium and cultivated for 24 h [18]. Then, a 2.0-mL culture with an OD600 value of approximately 2.5 was transferred into a 200-mL PVK medium (D-glucose, 10 g; (NH_4_)_2_SO_4_, 0.5 g; MgSO_4_·7H_2_O, 0.3 g; NaCl, 0.3 g; KCl, 0.3 g; FeSO_4_·7H_2_O, 0.03 g; MnSO_4_·4H_2_O, 0.03 g; Ca_3_(PO_4_)_2_, 10 g; and distilled water, 1000 mL, pH 7.0) [19] and cultured at 25 °C at 160 rpm for 9 d. Every day, a 5.0-mL culture was sampled and centrifuged at 3500 rpm for 15 min, and the supernatant was filtered through a 0.45-µm filter. The optical density was measured by the ammonium molybdate spectrophotometric method at 700 nm using an SQ-4802 UV/VIS spectrophotometer, and the WSIP content was calculated [20]. The pH of the mixture was measured using a pHS-3C acidity meter. The culture with no bacterial inoculation was used as a control (CK), and each treatment was repeated three times. The P-solubilizing amount was the WSIP of the sample minus the WSIP of control.

#### 2.4.3. Microcosm Setup and P Release by IPB Strains

The fresh sediments collected from the five sampling sites were mixed together and irradiated with a Co-60 γ-ray at a dose rate of 20 kGγ/h for 12 consecutive hours to kill all the microorganisms in the sediments. For each of the four representing IPB strains, 200 g of sterilized sediments were placed into a 1.0-L reagent bottle, followed by 800 mL of the filtered water for overlaying and 10 mL of the target IPB strain suspension with an OD value of approximately 2.7 added, and then covered with aluminum foil. Simultaneously, a control without adding an IPB strain suspension was set up. Finally, there were a total of five groups and with 15 repeats in each group. The samples were then cultured in an incubator at 13 °C (the mud–water interface temperature of Sancha Lake) at 80 rpm for 12 days, with manual shaking twice a day. During the cultivation, three bottles of each group were taken out on the 0th, 3th, 6th, 9th, and 12th days and NH_4_Cl (at a final concentration of 0.8 M after mixing) was added and shaken for 30 min to desorb the orthophosphate in the sediment. After centrifugation at 3500 rpm for 15 min, HCl-P and NaOH-P in sediments were extracted by the SMT method [21] and the supernatant was filtered with a 0.45-µm filter. The ammonium molybdate spectrophotometric method was used to determine the phosphorus contents in the sediments as well as in the supernatant [20].

#### 2.4.4. Statistical Analysis

A statistical analysis was performed using SPSS statistical software (version 20.0, IBM, Armonk, NY, USA). The one-way analysis of variance (ANOVA) was used to analyze the differences of HD/CD, WSIP, HCL-P, pH, and NaOH-P among different IPB strains. Spearsman correlation coefficients between soluble P released by bacteria and pH were calculated. Significance levels were set at *p* = 0.05 in all statistical analyses.

## 3. Results and Discussion

### 3.1. Results of IPB Screening and Identification

The online BLAST results of 16S rRNA gene sequences of the culturable IPB strains from the sediments of Sancha Lake are shown in Table 1. The strains sharing a similarity of >97% for the 16S rRNA gene sequences were considered belonging to the same species and that with a similarity of >95% were considered members of the same genus [22]. The isolated IPB strains with high 16S rDNA sequences similarities (same strain number as the nearest phylogenetic neighbor) were compared directly using the DNAman software and were treated as clones of the same strain if there were no more than three base differences [23]. The Blast results suggested that the 43 isolated strains belonged to 22 different species. The 16S rRNA of SWSI1719 shared a 94.15% similarity with the *Micromonospora halophytica* strain DSM 43171 (jgi.1058864), suggesting that it might be a potentially new species of the *Micromonosporaceae* family. The 16S rRNA gene sequences of SWSI1728 exhibited a 96.2% similarity with the *Bacillus paramycoides* strain NH24A2 (MAOI01000012), and that of SWSI1734 exhibited a 95.9% similarity with the *Bacillus idriensis* strain SMC 4352-2 (AY904033), thus, suggesting that they are potentially two new IPB species of the genus *Bacillus*.

According to the results of the 16S rRNA gene sequence analyses (Figure 2), the 43 IPB strains belonged to the three phyla of the domain Bacteria (*Actinobacteria, Firmicutes, and Proteobacteria*), eight families (*Aeromonadaceae, Bacillaceae, Gordonia, Microbacteriaceae, Paenibacillaceae, Planococcaceae, Pseudomonadaceae, and Rhodococcus*), ten genera, and twenty two species (Figure 2 and Table 1). Among the 43 obtained strains, 33 belonged to the phylum *Firmicutes*, five belonged to the phylum *Proteobacteria*, and five belonged to the phylum *Actinobacteria*. The most IPB strains belonged to genus *Bacillus*, followed by *Paenibacillus*, while each of *Rhodococcus and Caryophanon* had only one strain of IPB. Previous studies have shown that *Bacillus, Pseudomonas, Gordonia, and Paenibacillus* are all common IPB genera [24]. We also observed that *Bacillus and Paenibacillus* were the dominant strains obtained in this study exhibiting a stable P-solubilizing effect, while the isolated *Rhodococcus* and *Caryophanon* genera were rarely reported as IPB.

### 3.2. P-Solubilizing Ability

#### 3.2.1. P-Solubilizing Activity of Screened IPB Strains

The HD/CD values of the 43 IPB strains are shown in Table 2. Among them, colonies of five strains showed no halo zones on the NBRIP-BPB plate. Eight strains demonstrated small halo zones, and their HD/CD values were especially less than 1.5. The HD/CD values of 30 strains were all greater than 1.5. According to the HD/CD values [17], among the 43 strains, five strains demonstrated no ability, 8 strains showed weak ability, and 30 strains showed strong ability for P solubilization. Thirty-eight strains can produce Phosphate-solubilizing halo, and SWWI172 had the biggest halo zone, with an HD/CD value of 4.7, which is significantly higher than other strains except the strain SWSI1725. The strains SWSI1713 and SWSI1732 had the smallest halo zone, with an HD/CD value of 1.1, which was significantly lower than other strains except SWSI171, SWSI173, SWSI1714, SWSI1721, SWWI179, and SWWI178.

As demonstrated by Zhiguang Liu et al. [25], not all acid-producing IPB produce a halo zone. Therefore, the screened 43 IPB strains were cultured in a shake flask in PVK medium, and their respective P-solubilizing abilities were determined. Although we tested all 43 strains in the sediments, to avoid complexity in presentation, the WSIP levels on various days of incubation are presented here for eight representative strains from the sediments (Figure 3). The variation curves of WSIP content in the supernatant of the flasks were not observed to be the same. The P-solubilizing ability and the time to reach a maximum WSIP content were different for each strain. Although, a common feature was that they all declined and tended to stabilize after reaching the maximum. This is because a part of the orthophosphate released by IPB was absorbed by the cells and the other part was released into the supernatant. The P-solubilizing ability of IPB can be regarded as a result of cell proliferation and increased phosphorus content in the supernatant. Therefore, when the WSIP content of the supernatant reached a maximum value, the phosphorus absorption by IPB for growth caused the decline in supernatant WSIP content which then attained equilibrium. The IPB obtained from Xi Lake sediments by Qian Yinchao Y et al. also had the same properties [9].

The maximum WSIP value and the corresponding culture time and pH of the 43 IPB strains are shown in Table 2. As shown, the WSIP values in the supernatants harboring IPB strains were all significantly higher than those of the control (*p* < 0.05). The abilities of most IPB strains to dissolve calcium phosphate varied significantly (*p* < 0.05). All liquid media with insoluble tricalcium phosphate turned from turbid to clear after IPB cultivation. After calibration with the control, the phosphorus content in the culture medium was in a range of 1.79–102.77 mg/L. The strain SWSI1725, belonging to genus Bacillus, had the best P-solubilizing effect, with 103.57 mg/L of dissolved phosphorus and a phosphorus increase of 102.77 mg/L compared with the control. The weakest P-solubilization effect was observed in SWSI1717, with dissolved phosphorus of only 2.66 mg/L, 1.79 mg/L higher than the control. In general, these results were consistent with the abovementioned halo-zone experiments but with slight differences, which may be due to the non-diffusion or unclear diffusion of the acid produced by the strains on the solid plates or due to the difference in ingredients between the NBRIP-BPB plates and the PVK liquid medium. Compared with the IPB strains isolated from soils by Zhiguang Liu [25], the IPB strains isolated from the sediments of Sancha Lake had a high diversity but with slightly lower P-solubilizing abilities. Contrarily, compared with the IPB strains isolated from shallow lake sediments by Qian Yinchao et al. [9], the IPB strains from this study possessed both a higher diversity and better P-solubilizing abilities.

In the PVK medium, compared with the control group, the pH of all the strains was found to decline, indicating that acidic substances were produced during the culture of IPB. The pH value of the culture of SWSI178 strain was different from that of the control although not significant. The pH values of the cultures of other 42 strains were significantly different from that of the control (*p* < 0.05). The pH values of the cultures of some IPB strains varied significantly (*p* < 0.05). According to the Spearman’s correlation analysis, there was a significant correlation between the IPB culture pH and the amount of phosphorus released (*r* = −0.803). Nevertheless, a lower pH did not mean a stronger ability to dissolve phosphorus. For example, the strain SWSI174 of genus Bacillus showed a low phosphorus content in the culture even when the pH was low. The microbial ability to solubilize P has led to several hypotheses. While Lin et al. [26] also revealed a significant correlation between the culture pH and the P-solubilizing ability, Asea reported a lack of correlation or only a weak correlation between pH and the phosphate release [27]. In another report, Bianco et al. [28] hypothesized that the P-solubilizing effect of phosphate-solubilizing microorganisms was caused by various organic acids produced by them that chelated with Ca^2+^, Fe^3+^, Fe^2+^, and Al^3+^ ions to dissolve the insoluble phosphate. Likewise, Bagyaraj et al. [29] proposed that the production of organic acids only partly contributed to phosphorus solubilization, while proton extrusion was another important mechanism. Therefore, the mechanism of phosphorus solubilization by microorganisms needs to be investigated further.

#### 3.2.2. P Release by IPB Strains in Sediments

Four representative strains with high activities were selected from the 43 IPB strains to conduct the microcosm experiment, of which the results are shown in Figure 4, Figure 5 and Figure 6. As can be seen from Figure 4, the WSIP content in the overlying water was significantly increased in the sterilized sediments on days 3, 6, 9, and 12 compared to that of the CK group (*p* < 0.05). The WSIP contents in the overlying water inoculated with various IPB strains differed at the same time point. For instance, on day 6, the WSIP content in the overlying water inoculated with SWSI1725 was significantly different from that of the other groups (*p* < 0.05). Likewise, on day 9, the WSIP content in the overlying water inoculated with SWSI1730 was significantly different from that of the other groups (*p* < 0.05).

As seen from Figure 4, after 6 days of cultivation, the WSIP content of the overlying water in the CK group reached an equilibrium of 0.17 mg/L and it was stabilized until day 9. This may be because the overlying water used for cultivation was filtered lake water and the mass concentration difference between the overlying water and the interstitial water was small, therefore leading to only a short equilibrium period. After inoculation with SWSI1725, SWSI1730, SWSI1713, and SWSI1720, the highest WSIP contents of the overlying water were increased by 10.6, 9.5, 7.4, and 4.5 folds, respectively, and it occurred on day 6 for the SWSI1725 group and on day 9 for the other groups. This result indicated that the inoculated strains in the sterilized system significantly promoted the decomposition and release of phosphorus although their sediment P-solubilizing abilities varied. Similar results were obtained by You Xuejing et al. [30] in the study of phosphorus release from sediments. After the four strains were inoculated, the WSIP contents increased on day 3, reached the maximum on day 6 or 9, and dropped on day 12. This may be because the acid produced by microorganisms promoted the release of phosphorus from the sediments; in the later stage, bacteria attained high cellular densities and required significant amounts of P to sustain their growth, and parts of solubilized P were incorporated into their biomass [31].

Figure 5 shows that, in the sterilized sediments, the HCL-P content in the sediments decreased significantly on days 3, 6, 9, and 12 compared to that of the CK group (*p* < 0.05). The HCL-P content in the sediments inoculated with various IPB strains differed at the same time point. For instance, on day 6, the HCL-P content in the sediments inoculated with SWSI1725 was significantly different from that of the other IPB inoculation groups (*p* < 0.05). Likewise, on day 9, the HCL-P content in the sediments inoculated with SWSI1730 was significantly different from that of the other groups (*p* < 0.05). After inoculation with SWSI1725, SWSI1730, SWSI1713, and SWSI1720, the HCL-P contents in the sediments were reduced by 28.3%, 20.3%, 24.8%, and 21.3%, respectively. The four strains had different HCL-P-solubilizing abilities, with SWSI1725 exhibiting the best ability. The tendency of WSIP dynamics was similar to that of the HCL-P content in the sediments.

The change in HCl-P content was not noteworthy under the physicochemical process alone, while the IPB significantly promoted the decomposition and release of HCl-P. Halder et al. [32] revealed that some microorganisms may dissolve HCl-P to release phosphorus by synthesizing and decomposing organic acids. In recent years, microorganisms with such functions and various organic acids have been found to exist widely in soil and sediment environments. According to the above theory, in the sterilized group of the sterilized system, due to the elimination of microorganisms, there would be no biodegradation at the water–sediment interface. Therefore, HCl-P is hardly decomposed and transformed. On the contrary, after inoculating IPB into the sediments in the non-sterilized and sterilized systems, a good growth of IPB indicates a highly active organic acid synthesis and phosphorus decomposition, resulting in the decomposition and release of a large amount of HCl-P. Nevertheless, while this is inconsistent with Huang Yanlin’s conclusion [33] that “HCl-P was a relatively inactive form of phosphorus in sediments and was not easily released”, it is consistent with Nilanjan Maitra’s conclusion [4] that “HCl-P in sediments was an active form of phosphorus that could be dissolved by the organic acids secreted by microorganisms and released into the environment”.

Figure 6 shows that the NaOH-P content in the sediments decreased insignificantly on days 3, 6, 9, and 12 compared to that of the CK group (*p* < 0.05). The primary representatives of NaOH-P are the compounds of phosphorus bound to metals such as iron and aluminum and their oxides, and these metal compounds are easily displaced by the OH- produced by biological or physicochemical reactions to release orthophosphate into water bodies [34]. Moreover, the conversion of insoluble ferric iron to soluble ferrous iron under anoxic conditions could also lead to the release of orthophosphate [35]. In this study, we mimicked the lake environment in sterilized systems at neutral pH and high dissolved oxygen. The sediment NaOH-P contents were low in the sterilized groups and barely varied before and after the cultivation, indicating a minimal effect on the decomposition and release of NaOH-P by the IPB strains isolated and screened from the sediments of Sancha Lake. Our result was similar to the report of Fankem. Fankem found that IPB in soil had a great effect on HCL-P dissolving and had little effect on NaOH-P dissolving [36].

## 4. Conclusions

A total of 43 IPB strains were isolated and screened from the sediments of Sancha Lake. They were divided into three phyla, eight families, and ten genera as per the 97% similarity criterion of the 16S rRNA gene sequences. The dominant genera observed were *Bacillus* and *Paenibacillus*, while *Rhodococcus* and *Caryophanon* were rarely reported as IPB. The potentially new IPB species are SWSI1719, SWSI1728, and SWSI1734, indicating that the IPB strains screened from the sediments of Sancha Lake expand the range of P-solubilizing microorganisms.

The P-solubilizing experiments on 43 IPB strains on the NBRIP-BPB plates and in the PVK liquid media demonstrated that various IPB strains had significant differences in their P-solubilization abilities and that there was a significant correlation between the culture pH and the amount of phosphorus released. Some IPB strains exhibited no halo zone on the plate but showed a P-solubilizing ability when cultured in the liquid medium. The halo zone experiment on the plates cannot fully describe the ability of IPB to dissolve tricalcium phosphate. The microcosm results of four representative IPB strains inoculated into sediments showed that IPB strains significantly promoted the decomposition and release of HCl-P in sediments but had no significant effect on NaOH-P and that the acid production may be its primary P-solubilizing mechanism. The IPB in the sediments of Sancha Lake may be an important approach to provide phosphorus to eutrophic algae.

## Figures and Tables

**Figure 1 ijerph-16-02141-f001:**
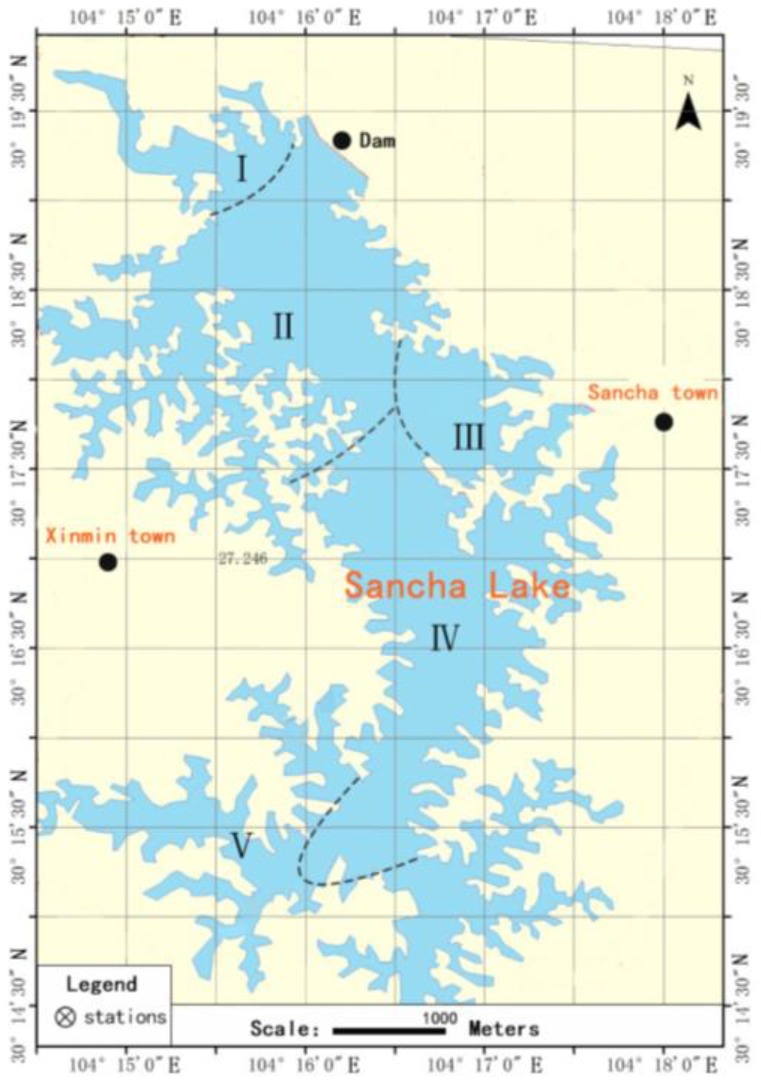
Sampling sites and function division in Sancha Lake.

**Figure 2 ijerph-16-02141-f002:**
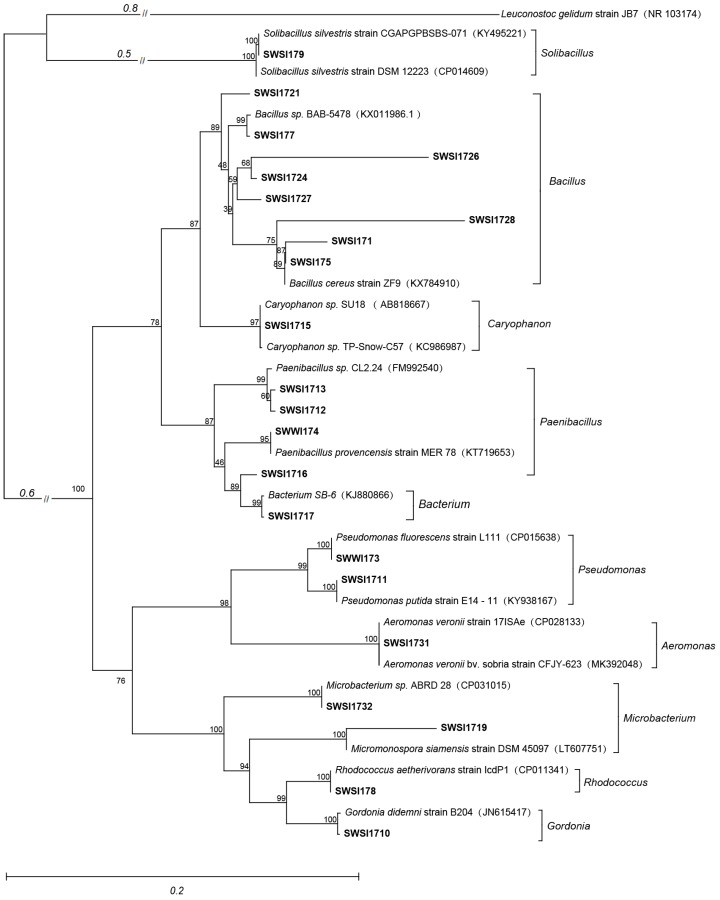
A neighbor-joining tree shows the phylogenetic relationships among 16S rDNA sequences of IPB and their closely related sequences from EzBioCloud. The numbers at the nodes indicate the bootstrap values based on the neighbor-joining analyses of 1000 resample data sets. The scale bar indicates evolutionary distance.

**Figure 3 ijerph-16-02141-f003:**
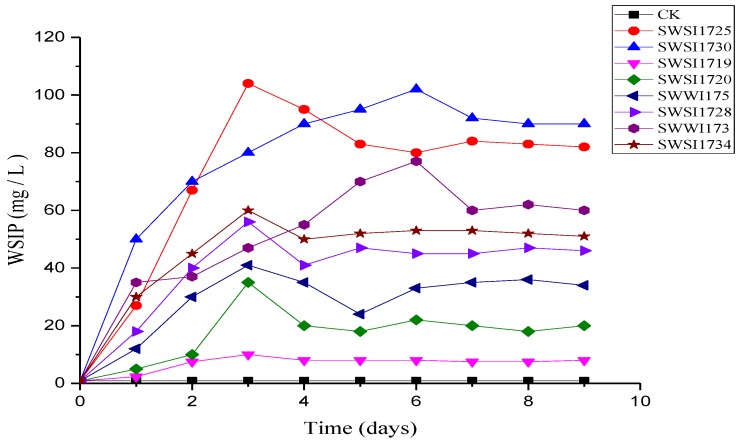
The Phosphate-solubilizing activities of 8 representative isolates cultured in PVK medium. Note: growing broth at 25 °C after 1, 2, 3, 4, 5, 6, 7, 8, and 9 days of growth. Each value is a mean of 3 independent replicates. CK: uninoculated control. The strains of SWSI1725, SWSI1730, SWSI1719, SWSI1720, SWWI175, SWWI173, SWSI1728, and SWSI1734 indicate the treatments inoculated, respectively.

**Figure 4 ijerph-16-02141-f004:**
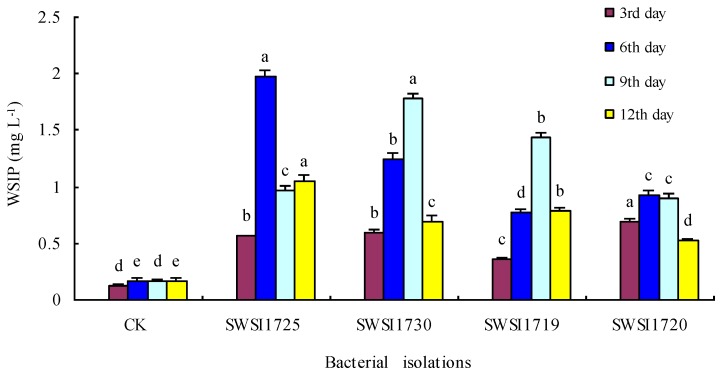
Comparative increases or decreases in WSIP by representative bacterial isolates in Sterile sediments on different incubation days in overlying water. Note: The strains of SWSI1725, SWSI1730, SWSI1719, and SWSI1720 indicate the treatments inoculated. Data are means ± S.E., with the same letters on the same day of incubation denoting insignificant differences among bacterial isolates.

**Figure 5 ijerph-16-02141-f005:**
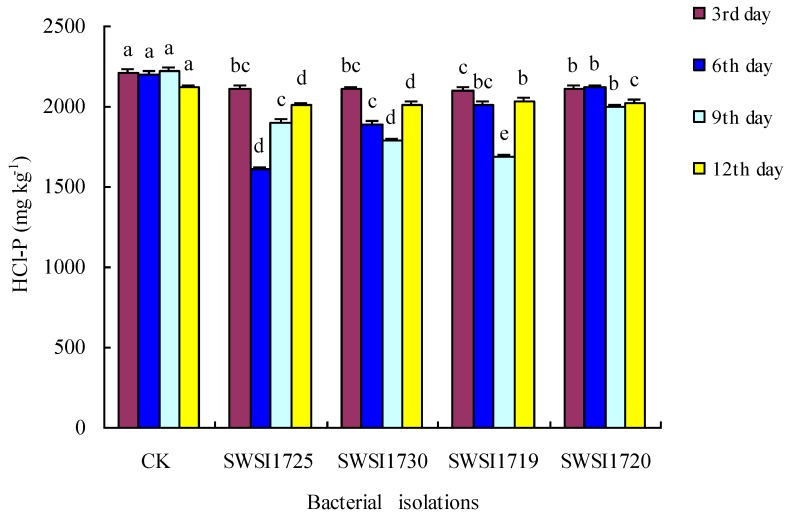
Comparative increases or decreases in HCl-P by representative bacterial isolates in Sterile sediments on different incubation days. Note: The strains of SWSI1725, SWSI1730, SWSI1719, and SWSI1720 indicate the treatments inoculated. Data are means ± S.E., with the same letters on the same day of incubation among bacterial isolates denoting insignificant differences.

**Figure 6 ijerph-16-02141-f006:**
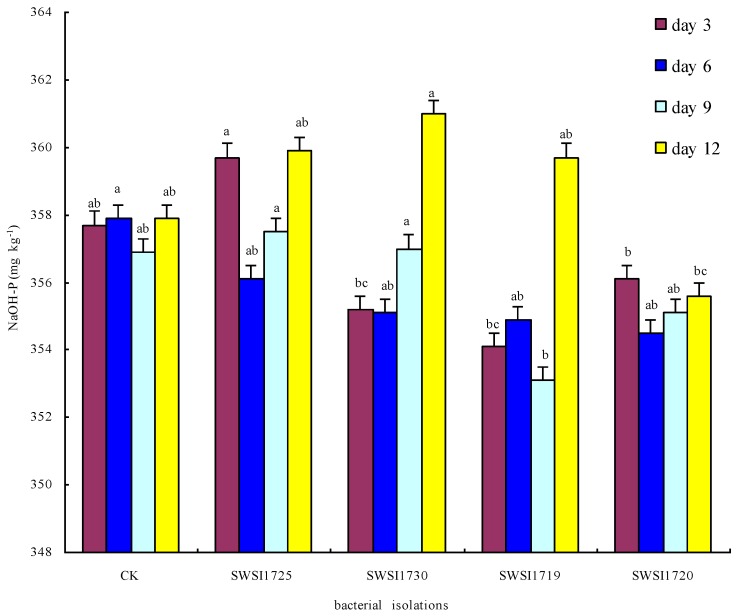
Comparative increases or decreases in NaOH-P by representative bacterial isolates in Sterile sediments on different incubation days. Note: The strains of SWSI1725, SWSI1730, SWSI1719, and SWSI1720 indicate the treatments inoculated. Data are means ± S.E., with the same letters on the same day of incubation among bacterial isolates denoting insignificant differences.

**Table 1 ijerph-16-02141-t001:** The online BLAST results of 16S rRNA gene sequences for the 43 inorganic phosphate-solubilizing bacteria (IPB) strains in the sediments of Sancha Lake.

Bacteria Strain	Nearest Phylogenetic Neighbor (Accession Number) ^a^	Gene Identity (%) ^b^	Taxonomical Assignment	Accession Number ^c^
SWSI171	*Bacillus cereus strain* ATCC 14579 (AE016877)	98.3%	*Bacillus cereus*	MK569699
SWSI172	*Bacillus indicus* strain LMG 22858 (JGVU01000003)	99.01%	*Bacillus indicus*	MK569700
SWSI173	*Bacillus idriensis* strain SMC 4352-2 (AY904033)	100%	*Bacillus idriensis*	MK569701
SWSI174	*Bacillus cibi* strain DSM 16189 (JNVC01000024)	98.61%	*Bacillus idriensis*	MK569702
SWSI175	*Bacillus proteolyticus* strain TD42 (MACH01000033)	100%	*Bacillus proteolyticus*	MK569703
SWSI177	*Bacillus aryabhattai* strain B8W22 (EF114313)	100%	*Bacillus aryabhattai*	MK569705
SWSI1714	*Bacillus proteolyticus* strain TD42 (MACH01000033)	100%	*Bacillus proteolyticus*	MK569712
SWSI1718	*Bacillus idriensis* strain SMC 4352-2 (AY904033)	100%	*Bacillus idriensis*	MK569716
SWSI1721	*Bacillus firmus* strain NBRC 15306 (BCUY01000205)	99.6%	*Bacillus firmus*	MK569719
SWSI1722	*Bacillus idriensis* strain SMC 4352-2 (AY904033)	100%	*Bacillus idriensis*	MK569720
SWSI1723	*Bacillus flexus* strain NBRC 15715 (BCVD01000224)	99%	*Bacillus flexus*	MK569721
SWSI1724	*Bacillus idriensis* strain SMC 4352-2 (AY904033)	99%	*Bacillus idriensis*	MK569722
SWSI1725	*Bacillus idriensis* strain SMC 4352-2 (AY904033)	100%	*Bacillus idriensis*	MK569723
SWSI1726	*Bacillus indicus* strain LMG 22858 (JGVU01000003)	98.61%	*Bacillus indicus*	MK569724
SWSI1727	*Bacillus pumilus* Strain FJAT-21963 (LJIY01000004)	100%	*Bacillus pumilus*	MK569725
SWSI1728	*Bacillus paramycoides* strain NH24A2 (MAOI01000012)	96.2%	*Bacillus sancha* sp.nov.	MK569726
SWSI1729	*Bacillus idriensis* SMC 4352-2 (AY904033)	100%	*Bacillus idriensis*	MK569727
SWSI1734	*Bacillus idriensis* strain SMC 4352-2 (AY904033)	95.9%	*Bacillus jianyang* sp.nov.	MK559751
SWWI171	*Bacillus firmus* strain NBRC 15306 (BCUY01000205)	100%	*Bacillus firmus*	MK569731
SWWI172	*Bacillus idriensis* strain SMC 4352-2 (AY904033)	99%	*Bacillus idriensis*	MK569732
SWWI175	*Bacillus idriensis* strain SMC 4352-2 (AY904033)	99%	*Bacillus idriensis*	MK569735
SWWI177	*Bacillus altitudinis* strain 41KF2b (ASJC01000029)	100%	*Bacillus altitudinis*	MK569737
SWWI179	*Bacillus idriensis* SMC 4352-2 (AY904033)	99%	*Bacillus idriensis*	MK569739
SWSI1716	*Paenibacillus populi* strain LAM0705 (KJ000069)	99%	*Paenibacillus populi*	MK569714
SWSI1712	*Paenibacillus* sp. strain FSL A5-0031 (MRTD01000002)	99.3%	*Paenibacillus* sp.	MK569710
SWSI1713	*Paenibacillus sp*. strain FSL A5-0031 (MRTD01000002)	99%	*Paenibacillus* sp.	MK569711
SWSI1720	*Paenibacillus agaridevorans* strain DSM 1355 (AJ345023)	99%	*Paenibacillus agaridevorans*	MK569718
SWWI174	*Paenibacillus provencensis* strain 4401170 (EF212893)	100%	*Paenibacillus provencensis*	MK569734
SWWI178	*Paenibacillus provencensis* strain 4401170 (EF212893)	100%	*Paenibacillus provencensis*	MK569738
SWSI1717	*Paenibacillus* sp. strain AZH4 (EU592044)	99%	*Paenibacillus sp.*	MK569715
SWSI1711	*Pseudomonas asiatica* strain RYU5 (MH517510)	100%	*Pseudomonas asiatica*	MK569709
SWSI1730	*Pseudomonas* sp. Strain TKP (CP006852)	100%	*Pseudomonas* sp.	MK569728
SWWI173	*Pseudomonas* sp. Strain TKP (CP006852)	98.3%	*Pseudomonas* sp.	MK569733
SWSI176	*Solibacillus silvestris* strain DSM 12223 (CP014609)	99%	*Solibacillus silvestris*	MK569704
SWSI179	*Solibacillus isronensis* strain B3W22 (AMCK01000046)	99%	*Solibacillus isronensis*	MK569707
SWSI1710	*Gordonia terrae* strain NBRC 100016 (BAFD01000032)	100%	*Gordonia terrae*	MK569708
SWWI176	*Gordonia hongkongensis* strain HKU50 (LC072670)	100%	*Gordonia hongkongensis*	MK569736
SWSI1731	*Aeromonas ichthiosmia* strain DSM 6393(X71120)	100%	*Aeromonas ichthiosmia*	MK569729
SWSI1733	*Aeromonas veronii* strain CECT 4257 (CDDK01000015)	99%	*Aeromonas veronii*	MK562701
SWSI1719	*Micromonospora halophytica* DSM 43171 (jgi.1058864)	94.2%	*Micromonosporaceae jiansan* sp.nov.	MK569717
SWSI1732	*Microbacterium invictum* strain DC-200 (AM949677)	98.74%	*Microbacterium invictum*	MK569730
SWSI178	*Rhodococcus aetherivorans* strain 10bc312 (AF447391)	99%	*Rhodococcus aetherivorans*	MK569706
SWSI1715	*Caryophanon tenue* strain DSM 14152(T) (MASJ01000025)	97.39%	*Caryophanon tenue*	MK569713

Note: ^a^ The sequence with highest percentage of identity observed in EzBioCloud; ^b^ The percentage of identity with EzBioCloud analysis; ^c^ The accession number of IPB in NCBI.

**Table 2 ijerph-16-02141-t002:** The Phosphate-solubilizing halo, maximum amounts of P-liberated, pH, and time of growth.

Bacteria Strain	Phosphate-Solubilizing Halo (HD/CD) ^a^	P-Liberated (mgL^−1^) ^b^	Time of Growth (h) ^c^	pH ^d^
CK		0.87 ± 0.09		7.03 ± 0.06 a
SWSI171	1.2 ± 0.1 h	4.37 ± 0.34 j	72	4.1 ± 0.1 f
SWSI172	1.9 ± 0.2 fg	5.41 ± 0.42 j	72	6.2 ± 0.1 b
SWSI173	1.5 ± 0.2 gh	4.02 ± 0.37 j	72	5.1 ± 0.2 d
SWSI174	1.8 ± 0.2 fg	5.09 ± 0.44 j	72	3.9 ± 0.1 f
SWSI175	2.7 ± 0.3 e	17.02 ± 1.34 gh	72	3.6 ± 0.0 g
SWSI177	2.1 ± 0.3 ef	10.703 ± 1.03 i	72	4.6 ± 0.2 e
SWSI1714	1.4 ± 0.2 gh	4.092 ± 0.44 j	72	5.0 ± 0.1 d
SWSI1718	1.9 ± 0.3 fg	4.86 ± 0.39 j	72	5.3 ± 0.2 d
SWSI1721	1.2 ± 0.1 h	4.65 ± 0.49 j	72	5.1 ± 0.3 d
SWSI1722	2.9 ± 0.3 de	19.72 ± 2.49 g	72	3.7 ± 0.2 g
SWSI1723	2.5 ± 0.3 ef	15.72 ± 1.43 h	72	3.6 ± 0.1 g
SWSI1724	3.5 ± 0.4 cd	21.15 ± 1.76 g	72	3.9 ± 0.2 f
SWSI1725	4.3 ± 0.5 ab	102.77 ± 4.06 a	72	3.1 ± 0.1 h
SWSI1726	2.9 ± 0.3 de	21.25 ± 2.70 g	72	3.9 ± 0.2 f
SWSI1727	2.3 ± 0.1 ef	13.84 ± 1.00 hi	72	3.8 ± 0.3 f
SWSI1728	2.8 ± 0.3 e	55.27 ± 3.72 d	72	3.4 ± 0.1 g
SWSI1729	1.8 ± 0.2 fg	17.51 ± 1.02 gh	72	3.9 ± 0.2f
SWSI1734	4.1 ± 0.3 bc	59.32 ± 2.71 d	72	3.5 ± 0.4 g
SWWI171	2.6 ± 0.3 ef	15.71 ± 1.70 h	72	3.7 ± 0.1 g
SWWI172	4.7 ± 0.5 a	82.43 ± 0.70 c	72	3.2 ± 0.1 h
SWWI175	3.0 ± 0.4 de	41.10 ± 0.42 e	72	3.4 ± 0.1 g
SWWI177	2.1 ± 0.1 ef	12.31 ± 0.82 i	72	3.7 ± 0.3 g
SWWI179	1.2 ± 0.1 h	2.95 ± 0.34 k	144	6.0 ± 0.2 b
SWSI1716	1.9 ± 0.3 fg	6.22 ± 0.71 j	72	5.9 ± 0.0 c
SWSI1712	1.6 ± 0.2 fg	5.08 ± 0.41 j	144	6.1 ± 0.1 b
SWSI1713	1.1 ± 0.1 h	4.85 ± 0.54 j	144	6.2 ± 0.2 b
SWSI1720	2.5 ± 0.2 ef	34.23 ± 1.91 f	72	4.9 ± 0.2 e
SWWI174	2.1 ± 0.3 ef	5.51 ± 0.56 j	144	6.1 ± 0.2 b
SWWI178	1.5 ± 0.1 gh	7.90 ± 0.65 j	144	5.1 ± 0.1 d
SWSI1711	2.2 ± 0.2 ef	12.27 ± 0.85 i	144	4.0 ± 0.3 f
SWSI1730	3.5 ± 0.4 cd	101.71 ± 5.25 a	144	3.2 ± 0.1 h
SWWI173	3.0 ± 0.3 de	76.91 ± 2.26 cd	144	4.4 ± 0.2 e
SWSI176	1.6 ± 0.2 fg	2.95 ± 0.23 k	144	5.8 ± 0.1 c
SWSI179	1.7 ± 0.1 fg	4.93 ± 0.36 j	48	6.0 ± 0.1 b
SWSI1710	ND	2.29 ± 0.16 k	48	6.5 ± 0.1 b
SWWI176	ND	3.22 ± 0.45 j	48	5.2 ± 0.1 d
SWSI1731	3.5 ± 0.3 cd	71.71 ± 3.15 cd	72	4.0 ± 0.1 f
SWSI1733	3.4 ± 0.4 cd	96.21 ± 7.05 b	72	4.0 ± 0.11 f
SWSI1719	2.3 ± 0.3 ef	8.37 ± 0.85 i	72	6.0 ± 0.1 b
SWSI1732	1.1 ± 0.2 h	3.35 ± 0.45 j	72	6.0 ± 0.3 b
SWSI178	ND	3.04 ± 0.49 j	72	6.7 ± 0.3 a
SWSI1715	ND	3.96 ± 0.42 j	72	6.4 ± 0.2 b
SWSI1717	ND	1.79 ± 0.36 k	72	6.4 ± 0.1 b

Note: Each value is represented by the mean ± S.E; control (CK): no inoculation; ND: not detectable; ^a^ HD/CD, the ratio of halo zone to colony diameter in the National Botanical Research Institute’s phosphate-bromophenol blue (NBRIP-BPB) medium; ^b^ the maximum water-soluble inorganic phosphorus (WSIP) value of the sample after subtracting the CK WSIP value in a PVK medium; ^c^ time of growth (h) in which maximum levels of soluble P were released in the PVK medium; ^d^ pH in which maximum levels of soluble P were released in the PVK medium. In the same column, data with different letters such as a, b and c indicate significant differences, while data with one or more same letters indicate insignificant differences at the 0.05 level. For example data with letters ab were insignificantly different from data with letters ab, a, b and bc. Data with letters bc were insignificantly different from data with letters bc, b, c and cd. Data with letters de were insignificantly different from data with letters de, d, e and ef. The rest can be done in the same manner.

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
