# Peer review of "Characteristics of Inorganic Phosphate-Solubilizing Bacteria from the Sediments of a Eutrophic Lake"

_ijerph, 2019, doi:10.3390/ijerph16122141_

Round 1

Reviewer 1 Report

The manuscript ijerph-506957 titled “Characteristics of inorganic phosphate-solubilizing bacteria from the sediments of a eutrophic lake” that I have referred dealts with isolation, characterisation and assessment of inorganic phosphate-solubilising bacteria from sediments of an eutrophic lake.  The manuscript fits well in the scope of the Journal but significant improvement (as detailed below) are required to be accepted. Based on this, I recommend to reconsider after major revision.

Overall the use of English must be improved through the text.

Latin names, i.e. species names, bacterial genera and families, in vitro, etc. must be in italics. Please apply this to all the sections, figures and tables.

Legends and headings must contain sufficient details to understand figures and tables away the text. Please apply this all of them.

Some references are not properly used or placed in the text. Please check if appropriated where and how they were used.

Abstract.

Specify which strains are considered novel species and in which genera.

Paraphrase the sentence “rich in IPB diversity”. The term “diversity” is not properly used.

Introduction

Please clarify the meaning of the sentence “In eutrophic water under effective control…”

Please include reference for SMT.

Please clarify the meaning of the sentence “ there is very limited research on the IPB species ..” species or strains?? Make more sense talking about strains. In the same way, clarify if the 89 species of pure culture IPB mentioned earlier in the text are not referring strains instead.

Reference after “…Sichuan Province”

Please reference [16] later in the sentence.

Please clarify aims, those don’t fit with what was carried out.

Material and methods

Section 2.1. needs references.

Please improve flow of the first three sentences of the second paragraph merging them.

Please justify the use of reference [17].

Section 2.2.

0.1 mL suspension was inoculated into …

Use superscripts for exponential numbers.

Please specify the number the number of rounds or give a minimum number instead.

Clarify the sentence starting “ the colonies with clear….”

Section 2.3. Improve structure of this section. Provide additional details about how contigs were made,how sequences were curated,  models used to infer the tree, number bootstraps, etc. Provide reference for MEGA software.

Section 2.4.1 Provide growth conditions for preparing the bacterial suspension. Number of replicates. Please clarify what was “carefully studied” when identified a clear halo zone or paraphrase. Please clarify the use of reference number [20].

Section 2.4.2

Please clarify why a different media were used to test phosphate solubilisation in broth. Provide details about the “pre-treated” process carried out with Excel. Provide details about how was determined the log phase of each bacteria and results.

2.4.3 Please clarify number of replicates used. It is not clear if there were 6 or just 1, as just one bottle was used to analyze. In the tables there are just single values. Clarify the use of reference {3}. Provide centrifugation details.

Statistical analysis section is missing.

Results and discussion

Overall discussion is weak. Deeper analysis using a wider range of references and considering the full picture is needed.

3.1 it is mainly material and methods.

3.2. Merge remaining content of 3.1 with 3.2. Please recall the use of the term “species”. It is not properly used.  If you need references to justify the use of a threshold at 97% as species delimitation. In fact currently other more strict threshold are more commonly used. Please see https://www.ncbi.nlm.nih.gov/pubmed/23591456

Check species delimitation as having 94% of similarity could better involve a novel genus instead.

Has the redundancy of strains been tested? How do you know you did not tested clones?

Table 1. Use validly named type species for taxonomic affiliation and determine the novelty of your strains. Possibly it is more appropriate to calculate 16S rRNA gene similarities using EzBioCloud as you have the option to calculate them just using validly named species.

Section 3.31. How isolates were “activate” (cultivated) is not included in material and methods. HD/CD < 1.5. In general flow of this section needs significant improvement. Significant part of results contained in table 1 and how those are related with the other parameter are not mentioned (possibly the use of a factorial ANOVA or similar approach would make sense). As mentioned in material and methods the use of a different medium to test solubilisation in broth is not justified. It is identified as a potential flaw in experimental design. No direct comparison can be done by using different media under different conditions.

Section 3.3.2 Clarify sentence 2 and 3. Please improve flow of section for better understanding. In Table 3, there is not applied statistical analysis mentioned in text. How many replicated were used? What numbers indicate? More details in heading needed.

References, Please check formatting style. Some given names were used as surnames (ref. 7, 8, 9, 10 and 14).

Author Response

Comments and Suggestions for Authors

1.     Overall the use of English must be improved through the text.

  Reply: Accepted and it was changed.

2.     Latin names, i.e. species names, bacterial genera and families, in vitro, etc. must be in italics. Please apply this to all the sections, figures and tables.

Reply: Accepted and it was changed.

3. Legends and headings must contain sufficient details to understand figures and tables away the text. Please apply this all of them.

Reply: Accepted and it was changed.

4.Some references are not properly used or placed in the text. Please check if appropriated where and how they were used.

Reply: Accepted and the related references were revised.

Abstract.

5.Specify which strains are considered novel species and in which genera.

Reply: Accepted and the related information is supplemented in the part of abstract and Table 1.

6.Paraphrase the sentence “rich in IPB diversity”. The term “diversity” is not properly used.

Reply: Accepted and it was changed.

Introduction

7.Please clarify the meaning of the sentence “In eutrophic water under effective control…

Reply: Accepted and it was changed :While assessing the exogenous phosphorus in effectively controlled eutrophic water.

8. Please include reference for SMT.

Reply: Accepted and the reference for SMT was included.

9.Please clarify the meaning of the sentence “ there is very limited research on the IPB species ..” species or strains?? Make more sense talking about strains. In the same way, clarify if the 89 species of pure culture IPB mentioned earlier in the text are not referring strains instead.

Reply: In the sentence: “ there is very limited research on the IPB species ..”, “species” meant strains, and revision was done. In the sentence mentioned earlier in the textUntil now, 39 genera and 89 species of…. , “species” refered species, not strains.

10. Reference after “…Sichuan Province”

Reply: Accepted and it was changed.

11.Please reference [16] later in the sentence.

Reply: Accepted and it was changed.

12.Please clarify aims, those don’t fit with what was carried out.

 Reply: Accepted and it was changed.

Material and methods

13. Section 2.1. needs references.

 Reply: Accepted and references was added.

14. Please improve flow of the first three sentences of the second paragraph merging them.

 Reply: Accepted and it was changed.

15. Please justify the use of reference [17].

Reply: Accepted and reference [17]was deleted.

Section 2.2.

16. 0.1 mL suspension was inoculated into …

 Reply: Accepted and it was changed.

17. Use superscripts for exponential numbers.

 Reply: Accepted and it was changed.

18.Please specify the number the number of rounds or give a minimum number instead.

 Reply: Accepted and it was changed.

19.Clarify the sentence starting “ the colonies with clear….”

    Reply: “ the colonies with clear….” was changed into The colonies, which had clear halo zones or were growing well….

20. Section 2.3. Improve structure of this section. Provide additional details about how contigs were made,how sequences were curated,  models used to infer the tree, number bootstraps, etc. Provide reference for MEGA software.

  Reply: Accepted and it was changed.

21. Section 2.4.1 Provide growth conditions for preparing the bacterial suspension. Number of replicates. Please clarify what was “carefully studied” when identified a clear halo zone or paraphrase. Please clarify the use of reference number [20].

 Reply: Accepted and it was changed.

Section 2.4.2

22. Please clarify why a different media were used to test phosphate solubilisation in broth. Provide details about the “pre-treated” process carried out with Excel. Provide details about how was determined the log phase of each bacteria and results.

Reply: culture time was 24 hours according to the report by Zhou et al.Identification and determination of growth curve of four bacterium isolated from Taihu lake .Journal of lake science.1998,10,4,60-62.and the log phase of each bacteria wasn’t determined.

I had thought it was better to use different media to test phosphate solubilization in broth, but no comparison was done for the test results of phosphate solubilization using different media. The potential flaw in experimental design was supplemented in the discussion.

There was something wrong with the expression “pre-treated process carried out with Excel” and it was changed into “the data was analyzed using SPSS 20.0”.

23. 2.4.3 Please clarify number of replicates used. It is not clear if there were 6 or just 1, as just one bottle was used to analyze. In the tables there are just single values. Clarify the use of reference {3}. Provide centrifugation details.

 Reply:

---Accepted and revision was done. The number of replicates was 3 and related information was supplemented in the text.

---Reference [3] was deleted and centrifugation details were supplemented in the text.

24.Statistical analysis section is missing.

  Reply: Accepted and statistical analysis was supplemented.

Results and discussion

25.Overall discussion is weak. Deeper analysis using a wider range of references and considering the full picture is needed.

Reply: Accepted and it was changed

263.2. Merge remaining content of 3.1 with 3.2. Please recall the use of the term “species”. It is not properly used.  If you need references to justify the use of a threshold at 97% as species delimitation. In fact currently other more strict threshold are more commonly used. Please see https://www.ncbi.nlm.nih.gov/pubmed/23591456

Reply: Accepted and it was changed.

27. Check species delimitation as having 94% of similarity could better involve a novel genus instead.

Reply: Two potentially new strains, SWSI1728 and SWSI1734 belonged to genus Bacillus, and a potentially new strain, SWSI1719 belonged to family Micromonosporaceae.

28.Has the redundancy of strains been tested? How do you know you did not tested clones?

Reply: The redundancy of strains was tested. Whether clones were tested was determined by comparison results using DNAman software and the related information was supplemented in the part of 3.1.

29.Table 1. Use validly named type species for taxonomic affiliation and determine the novelty of your strains. Possibly it is more appropriate to calculate 16S rRNA gene similarities using EzBioCloud as you have the option to calculate them just using validly named species.

 Reply: Accepted and it was changed.

30. Section 3.31. How isolates were “activate” (cultivated) is not included in material and methods. HD/CD 1.5. In general flow of this section needs significant improvement. Significant part of results contained in table 1 and how those are related with the other parameter are not mentioned (possibly the use of a factorial ANOVA or similar approach would make sense). As mentioned in material and methods the use of a different medium to test solubilisation in broth is not justified. It is identified as a potential flaw in experimental design. No direct comparison can be done by using different media under different conditions.

  Reply: Accepted and revision was done.

---The one-way analysis of variance (ANOVA) was used to analyze the differences of HD/CD, WSIP, HCL-P, pH and NaOH-P among different IPB strains.

---I had thought it was better to use different media to test phosphate solubilization in broth, but no comparison was done for the test results of phosphate solubilization using different media. The potential flaw in experimental design was supplemented in the discussion.

 31.Section 3.3.2 Clarify sentence 2 and 3. Please improve flow of section for better understanding. In Table 3, there is not applied statistical analysis mentioned in text. How many replicated were used? What numbers indicate? More details in heading needed.

  Reply: Accepted and it was changed. Statistical analysis was applied to the data in former Table 3 and relative information and figures were supplemented in the text.

32.References, Please check formatting style. Some given names were used as surnames (ref. 7, 8, 9, 10 and 14).

  Reply: Accepted and revision was done.

此页面预览技术

Reviewer 2 Report

The authors have screened and isolated IPB species from the sediments of Sancha Lake and investigated their ability to release inorganic P. Overall, the study is interesting and the experimental design was well-conceived. Still, I have some concerns regarding the lack of statistical treatment to some of the most important results. See specific comments below.

IMPORTANT: The manuscript is missing line numbers, which makes the reviewing process very difficult. Don't forget to add them in the upcoming revision.

Page 1, "According to the Standards Measurements and Testing Program of the European Commission (SMT), ...." -> needs a supporting reference!

Page 4, "The data was pre-treated with Excel and analyzed using SPASS 20.0." -> SPASS? Perhaps SPSS? If so, an additional section about statistical analysis should be added and specific details about the tests carried out should be provided.

Page 4, "After centrifugation, various forms of phosphorus in sediments were extracted by the SMT method [23]" -> Which forms? Please indicate the forms, as they are clearly a relevant part of the results.

Table 2: The results (solubilizing halo, P-liberated, and pH) should be subjected to a statistical comparison (ANOVA) to determine significant differences between the identified strains. Modify the Results and Discussion accordingly.

Table 3: Why are these results missing statistical treatment?

Author Response

Comments and Suggestions for Authors

1.IMPORTANT: The manuscript is missing line numbers, which makes the reviewing process very difficult. Don't forget to add them in the upcoming revision.

 Reply: Accepted and it was changed

2.Page 1, "According to the Standards Measurements and Testing Program of the European Commission (SMT), ...." -> needs a supporting reference!

 Reply: Accepted and it was changed

3.Page 4, "The data was pre-treated with Excel and analyzed using SPASS 20.0." -> SPASS? Perhaps SPSS? If so, an additional section about statistical analysis should be added and specific details about the tests carried out should be provided.

 Reply: Accepted and it was changed. The related information was supplemented in the text.

4.Page 4, "After centrifugation, various forms of phosphorus in sediments were extracted by the SMT method [23]" -> Which forms? Please indicate the forms, as they are clearly a relevant part of the results.

 Reply: Accepted and the related information was supplemented in the text.

5.Table 2: The results (solubilizing halo, P-liberated, and pH) should be subjected to a statistical comparison (ANOVA) to determine significant differences between the identified strains. Modify the Results and Discussion accordingly.

 Reply: Accepted and it was changed

6. Table 3: Why are these results missing statistical treatment?

Reply: Statistical treatment is supplemented for the data in former table 3. Three figures were added in the text.